# When most fMRI connectivity cannot be detected: Insights from time course reliability

**Jan Willem Koten**[1]*, **Hans Manner**[2], **Cyril Pernet**[3], **Andre Schüppen**[4], **Dénes Szücs**[5], **Guilherme Wood**[1], **John P. A. Ioannidis**[6]

1 Institute of Psychology, University of Graz, Graz, Austria, 2 Department of Economics, University of Graz, Graz, Austria, 3 Dept. Neurology and Neurobiology Research Unit, Copenhagen University Hospital, Copenhagen, Denmark, 4 IZKF—Interdisciplinary Center for Clinical Research, RWTH Aachen University, Aachen, Germany, 5 Department of Psychology, University of Cambridge, Cambridge, United Kingdom, 6 Department of Medicine, Department of Epidemiology and Population Health, Department of Biomedical Data Science, Department of Statistics, and Meta-Research Innovation Center at Stanford (METRICS), Stanford University, Stanford, California, United States of America

* jan.koten@uni-graz.at

**Data Availability Statement:** The data and software relevant to this study's results are publicly available from GitHub. The fMRI data and TrueCon software: https://github.com/hinata2305/TrueCon. The CleanBrain software: https://github.com/

## Abstract

The level of correlation between two phenomena is limited by the accuracy at which these phenomena are measured. Despite numerous group reliability studies, the strength of the fMRI connectivity that can be detected given the within-subject time course reliability remains elusive. Moreover, it is unclear how within-subject time course reliability limits the robust detection of connectivity on the group level. We estimated connectivity from a working memory task. The grand mean connectivity of the connectome equaled r = 0.41 (95% CI 0.31–0.50) for the test run and r = 0.40 (95% CI 0.29–0.49) for the retest run. The mean connectivity decreased to r = 0.09 (95% CI 0.03–0.16) when test-retest reliability and auto-correlations of single time courses were considered, indicating that less than a quarter of connectivity is detectable. The square root of the detectable connectivity r = 0.09 suggests that only 0.81% of the connectivity is explained by working memory-related time course fluctuations. Null hypothesis significance testing (NHST)-based analysis revealed that within-subject time course reliability markedly affects the significance levels at which paths can be detected at the group level. This was in particular the case when samples were small or connectome coordinates were randomly selected. With a sample of 50 individuals, the connectome of a test session was completely reproduced in a retest session at P < 2.54e$^{-6}$ despite the fact that almost no connectivity was explained by the cognitive experiment. Within-subject time course reliability can offer valuable insights on the detectable connectivity and should be assessed more frequently.

## Introduction

It is believed that the energy required to perform cognitive tasks in the brain is indirectly captured by the so-called blood oxygen level dependent (BOLD) response, which can be measured using functional magnetic resonance imaging (fMRI). It is reasonable to assume that the

hinata2305/CleanBrain. The data: https://github.com/hinata2305/Data.

**Funding:** This study was funded by the FWF grant P 22577-B18. Wood.

**Competing interests:** The authors have declared that no competing interests exist.

characteristics of task-related signals should be similar to each other when the same task is performed twice under identical measurement conditions within the same person and referred to as *Within-subject time course reliability*. Within-subject time course reliability is essential in assessing if fMRI is suitable as biomarker since medical imaging is inherently focused on the level of the single subject. Within-subject time course reliability differs markedly from mainstream group test-retest reliability measurers that are used in a scientific context [1–5]. Notably, within-subject time course reliability cannot be estimated from "resting state" data as test and retest time courses of the latter are inherently unsynchronized in nature. Currently, it is believed that two brain regions are connected with each other when they show synchronized behavior. The *connectivity* among two brain regions is usually obtained by correlating the time courses of the respective brain regions. The detectable connectivity that can be achieved in neuro-imaging is closely related to a phenomenon first described in the 1970s [6].

$$\text{correlation upper bound AB} = \sqrt{(\rho A * \rho B)}$$

Nunally argued that the correlation among two observations A and B cannot exceed the test-retest reliability of observations A and B. In which $\rho A$ and $\rho B$ denote the test-retest reliability of the two phenomena under study.

$$\text{connectivity upper bound AB} = \sqrt{(\rho A * \rho B)}$$

In analogy we argue that the connectivity upper bound among two brain regions A and B cannot exceed the within subject time course reliabilities of region A and B [6, 7]. In which $\rho A$ and $\rho B$ denote the test-retest reliability of the two time courses under study. If connectivity exceeds the connectivity upper bound, one might infer that it is overestimated and should be corrected by replacing the connectivity with the connectivity upper bound.

Spuriously high observed and reported correlations, also known as *"Voodoo correlations"*, have attracted a lot of attention over the last decade and according to Fiedler they are seen "everywhere" in neuroimaging studies and many other scientific investigations [7, 8]. However, little attention has been given specifically to estimating the detectable level of connectivity based on within-subject time course reliability which may offer valuable insights about the nature and extent of observed fMRI connectivity. The within-subject time course reliability of task-driven fMRI is likely lower than the group reliability estimates, and may at best reach a correlation of ~0.25 [1]. The root of 0.25 suggests that no more than 6.25% of the connectivity variance can be detected.

The correct estimation of connectivity and reliability correlations is complicated by the occurrence of noise. The fMRI time course is contaminated by unwanted signals that are mainly related to respiration, cardiac pulsations, neural and hemodynamic noise as well as scanner-induced low-frequency drifts and head motion [9]. Currently, attempts are being made to remove these noise factors by incorporating white matter ventricle and motion time courses into a regression model. Unfortunately, many unwanted noise signals remain in the fMRI time course despite denoising. They can be detected when the fMRI time course is shifted by one unit and correlated with itself. The correlations that arise from this operation are referred to as *lag 1 autocorrelation*. Lag 1 autocorrelations can lead to an overestimation of the correlation when two fMRI time courses are correlated with each other, and some measures must be taken to control for this. An established way is to estimate the lag-1 autocorrelation of the residual time course, which is obtained by regressing the two time courses of interest against each other [10]. Once estimated, this weight can be applied to the individual time course of interest by subtracting the weighted, lagged time course from the unlagged time course. A second approach is block bootstrapping [11, 12]. Here, confidence intervals of

correlations are estimated by generating a larger number of simulated time courses that exhibit auto-correlated behavior. This is done by extracting temporally connected stretches of the time courses randomly. The stretches of the time course are referred to as blocks. The simulated time courses are then generated by concatenating randomly selected blocks in an overlapping or non-overlapping fashion

Currently, group analysis is commonly performed by testing the connectivity correlations obtained from a sample of individuals for a specific path within the network against zero using null hypothesis significance testing (NHST). This statistical approach is applied individually to all connections to identify those that are significantly different from zero at the group level. P values linked to NHST are the most commonly used (and misused) method to defend the scientific credibility of a study. It is not unlikely that the p values of many group studies are over optimistic as the connectivity correlations that are entered into the group analysis are possibly over estimated given the constraints of within-subject time course reliability.

The upper limit of connectivity that can be detected at the single subject level may affect the significance level at which group effects can be reproduced by means of null hypothesis significance testing (NHST) [13]. A conservative approach is to consider that the results of group studies are reproducible when the p values of two independent observations pass a threshold of statistical significance [14]. Against this backdrop the larger of the two observed p values is often taken as the statistical threshold at which a phenomenon is deemed reproducible and referred to as *conjunction analysis* [14–16]. Classic conjunction analysis has been criticized for being overly conservative and partial solutions have been suggested [14]. The *Dice overlap* measure, originally developed for ecological purposes in the 1940s [17] has been applied in neuroimaging as an alternative way to estimate the reproducibility of an image [18, 19] and can be seen as an extension of classic conjunction analysis. It calculates the proportion of the network shared between two observations by doubling the count of commonly present elements and dividing it by the total count of elements at a given significance threshold.

Here we test four hypotheses regarding the detectability of connectivity. We conducted a working memory experiment on two separate occasions and extracted two working memory-related functional brain networks. These networks were based on brain activity maxima identified in a previous large-scale meta-analysis that examined the neural correlates of working memory [20]. First, we investigated the *effect of within-subject time course reliability on detectable connectivity*. We hypothesized that connectivity among regions will be higher than the within-subject time course reliability. When observed connectivity exceeds the precision of time course measurement, we replace the connectivity correlation with the within-subject time course reliability correlation and refer to this as detectable connectivity. Second, we investigated the *effect of within-subject time course reliability on null hypothesis significance testing (NHST) based group reproducibility*. We expected that the mean detectable connectivity may be substantially lower than mainstream connectivity. Therefore, we hypothesized that mainstream group connectivity statistics obtained by testing connectivity correlations against zero are likely inflated. Third, we investigated the *effects of sample size on group reproducibility*. It is well established that power to detect group effects increases with larger sample sizes. We hypothesized that the differences between group statistics based on observed versus detectable connectivity will decrease with larger samples. Fourth, we investigated the *effect of connectome selection on group reproducibility*. It is reasonable to assume that a hypothetical connectome based on working memory-related brain coordinates may contain more robust brain activity data and therefore yield more reliable outcomes compared to randomly generated connectomes.

## Materials and methods

### Ethics statement

All subjects gave their written informed consent to participate in research before the actual experiment took place. All investigations were conducted according to the principles expressed in the Declaration of Helsinki. The study was approved by the Ethics committee of the University of Graz under GZ 39/31/63 ex 2011/12.

### Participant and task selection

Great care was taken to include individuals believed to be representative of the general population, with detailed demographics described elsewhere [21]. The average age was 41.8 years with std 17.39 years which approaches the average age of Austria which is currently 43.2 years [22]. The recruitment of the subjects started at august 2012 and ended at February 2014. To ensure data accuracy, we included only individuals with head motions less than 3 mm while the average correct response rate of the sample was 99%. 50 individuals were selected out of 67 individuals. This screening procedure aimed to minimize the risk that within subject time courses was due to poor task performance or excessive head motion. More details are given in S1 Text in S1 File.

### Task

We used a working memory task integrated with a distractor task. Comprehensive details of this task and its conventional group test-retest reliability can be found elsewhere [21]. In brief, participants were required to determine whether a presented letter belonged to a previously memorized set of two letters. The encoding and recognition phase was disrupted by a number Stroop task, intentionally designed to divert individuals' attention away from retaining letters in memory. The task cycle was repeated 24 times, leading to 24 "Stroop" and 24 "memory" responses that were delivered in alternation. Each cycle was interrupted by a jittered pause lasting approximately 12 seconds. The participants underwent testing on two distinct occasions, with an inter-session pause of at least an hour. More details are given in S1 Text in S1 File.

### Task analysis

Initially, we determined if the overall behavioral response patterns of the entire group remained consistent across the test and retest runs. This analysis aimed to minimize the possibility that the lack of mean group reproducibility on the neural level stemmed from dissimilar behavioral response patterns between sessions. To achieve this, we calculated the mean response time per item for the entire group, resulting in a total of 48 averaged response times (24 Stroop times and 24 memory items) for each run. The test-retest reliability of these response time curves reflects the consistency of entire task cycles across both sessions at the group level. Subsequently, our objective was to ascertain whether within-subject time course reliability correlates with within subject response time reliability. The analogy of within-subject time course reliability is obtained when the 48 response times of the test run are correlated with the 48 response times of the retest run within individuals resulting in 50 correlations. Finally, we correlated the 50 test-retest reliability estimates of response times with the 50 mean within-subject time course reliability estimates. The mean within-subject time course reliability was obtained by averaging the 34-time course reliabilities per subject.

## fMRI

MRI scans were performed on a 3 T Siemens Magnetom Skyra (Siemens Medical Systems, Erlangen, Germany) equipped with a 32-channel head coil. Functional imaging data were measured with iso voxel resolution of 4mm with a 10% gap and a repetition time of 1.24 seconds, while structural imaging data were measured with an iso voxel resolution of 1mm. Functional data were brought into 2D FS average space using spherical alignment procedures as available in FreeSurfer [23]. The grey matter time courses were brought into a custom MATLAB environment and subsequently detrended, denoised and corrected for head motions within a GLM framework [24]. The noise regressors consisted of 5 "white matter" components and 5 "ventricle" components. The motion regressors included the first 2 principal components of the head motion data. We did not extract brain regions from brain activation maxima as this may induce circularity [7]. Instead, 34 MNI coordinates that are believed to be essential for working memory were taken from a meta-analysis and brought into 2D FS average space [20]. We created a circle of 8 mm in diameter around the coordinate of interest on the 2D representation of the brain and extracted the time courses of interest which were subsequently averaged. Further details of the fMRI techniques are given in the S1 Text in S1 File.

## Statistics

The statistics detailed below were performed in the TrueCon software package [25].

**Observed connectivity.** The observed connectivity was computed per individual per session. It is standard practice to estimate functional connectivity by means of Pearson's correlations [26]. In total, 561 paths were derived from the lower triangle of the 34 nodes * 34 nodes correlation matrix. This led to 561 paths * 50 subjects = 20850 correlations per session. We estimated the CI of a single path correlation through block bootstrapping (n = 10,000 bootstraps) (see below).

**Averaging connectivity and its confidence interval.** In neuroimaging, it is common practice to include both positive and negative connectivities when creating an image. However, when reporting mean connectivity in the context of descriptive statistics, it can be informative to estimate absolute connectivity as well. Both negative and positive connectivities are valid observations of connectivity and should be treated equally in the analysis. We considered a connectivity negative when the CI upper bound of the correlation was below zero. In this case the correlation and it's lower and upper bound intervals were made absolute before the actual averaging procedure took place. All time courses were of equal length meaning that the degrees of freedom that were involved when estimating CIs was invariant in all cases. A weighting procedure that is sometimes considered when averaging CIs is obsolete for this reason.

**Conventional group reliability.** We estimated the test-retest reliability of every single path at the level of the group by correlating the 50 path estimates obtained at the test session with the 50 path estimates obtained at the retest session resulting in 561 reliability estimates. Subsequently, the 561 test-retest reliability estimates were averaged. We used Pearson correlations for this purpose to remain consistent with the other reliability estimates in this study even if intra-class correlations are available as an alternative [2].

**Within subject time course reliability.** We estimated the test-retest reliability with its corresponding CI for every single time course (n = 488 data points) resulting in 34 nodes * 50 subjects = 1700 test-retest correlations that were accompanied with their corresponding CIs. The CIs of the within subject time course reliabilities were estimated through block bootstrapping. Details of this procedure can be found in the section "Bootstrapping on the time course level". More information on the rationale behind Pearson correlations instead of ICC is given in S1 Text in S1 File.

**Treatment of negative test-retest reliability and averaging of data.** Treatment of negative test-retest reliability and averaging of data is discussed in S2 Text in S1 File.

**Connectivity upper bound.** Although the correlation upper bound is well established in group imaging it has not yet been applied to single subject connectivity estimation. The connectivity upper bound was defined according to Nunnally as:

$$connectivity\ upper\ bound\ AB = \sqrt{\rho_{node\ A*}\ \rho_{node\ B}} \tag{1}$$

In which $\rho_{node\ A\ or}\ \rho_{node\ B}$ denote the test-retest reliability of the time course within the subject for node (region) A or B respectively [6]. We estimated the connectivity upper bound for every single path from the two within-subject time course reliability estimates available resulting in 561 paths * 50 subjects = 20850 correlations. Formula 1 implies that correlations are non-negative hence negative correlations were set at zero when they were entered into the formula.

**Detectable connectivity.** The detectable connectivity depends on the observed connectivity that may either lay below or above the connectivity upper bound. The connectivity upper bound as described in the section above should not be confused with the confidence upper bound interval of a connectivity correlation! Observed connectivity was replaced by the connectivity upper bound when observed connectivity exceeded the connectivity upper bound. One might argue that the replacement of the observed connectivity by the connectivity upper bound value is only justified when a statistically significant difference between the two correlations exists. Instead of relying on a testing procedure, we argue that it is more informative to estimate the detectable connectivity along with the corresponding CIs. This allows us to directly compare the CIs of the observed connectivity and detectable connectivity, which is the crucial comparison we are interested in. Additionally, we aimed to derive a singular statistical measure that integrates both reliability and connectivity concepts. This measure would then enable the creation of a comprehensive neuroimage that conveys detectable connectivity based on the reliability of the observed time courses. We tried to establish this new measure of connectivity following the procedure below. In theory detectable connectivity can differ for a test and test and retest run despite common connectivity upper bound threshold.

We calculated the detectable connectivity from the observed connectivity and the connectivity upper bound using the following procedure:

We set observed connectivity to zero when one or two nodes exhibit negative time course reliability. This is a necessary operation since given Formula (1) one cannot take the root out of a negative reliability.

Otherwise,

If the observed connectivity is a positive correlation: In this case we compared the absolute connectivity and absolute connectivity upper bound correlations and took the smaller of the two correlations.

Else, if the observed connectivity is negative: we compared the absolute connectivity and connectivity upper bound correlations and took the smaller of the two correlations and made the sign of the result negative.

We implemented block bootstrapping to estimate the CIs for the detectable connectivity. Block bootstrapping is particularly advantageous as it can effectively handle non-normally distributed data.

**Distribution percentiles and deviations.**   In total 1700 test-retest correlations and 20850 (detectable) connectivity correlations were obtained. We estimated the 2.5 and 97.5 percentiles and standard deviations of the respective distributions. Finally, we estimated the CI of the distribution mean and distribution standard deviations using conventional bootstrapping.

**Fisher z transform.**   For estimating means of correlations, standard deviations of correlations and means of confidence intervals, correlations were both forward and back-transformed, whereas for estimating t-statistics, correlations were only subjected to forward transformation.

**Residual autocorrelations.**   Residual autocorrelations within and between regions were removed using an AR(1) model [10]. More details about the effectiveness of this procedure can be found in S1 Text in S1 File.

**Conjunction analysis.**   The 561 paths * 50 subjects in the test and retest session underwent NHST resulting in 561 t-statistics for each session. This was done for conventional and detectable connectivity estimates performed with or without AR(1) correction. The smaller t statistic of the test and retest session was used for conjunction analysis [27]. We also used the largest p value of the two observations as a measure of conjunction since it is not sensitive to sample size and therefore suited for our Monte Carlo simulations that involved sample sizes that ranged from 10–50 individuals.

**NHST based image reproducibility.**   We estimated the group reproducibility of a functional connectome at a particular significance threshold that was obtained through NHST using the well-established Dice overlap measure.

$$Dice = 2*(Test \cap Retest)/(Test + Retest) \tag{2}$$

Within this context "Test∩Retest" represents the number of elements that survive conjunction analysis while "Test+Retest" represents the total number of elements present in the thresholded image. We investigated the entire exponential threshold space on the basis of $p = 0.05^{(n*0.1)}$ covering values from $p = 0.74$ up to $p = 2.9802e^{-33}$. In addition, we indicated the point that refers to the Bonferroni corrected p value which was found at $0.05/561 = 8.91266e^{-05}$. All paths met the critical Bonferroni corrected p value even after conjunction analysis, rendering more liberal methods like FDR corrections unnecessary. The Bonferroni thresholded data in our experiment automatically satisfies FDR correction criteria.

**Visualization.**   The divers connectomes obtained where visualized using the "VisualConnectome" MATLAB package [28].

**Bootstrapping on the time course level.**   The CI of correlations can be biased when time courses exhibit autocorrelations. Hence, the CIs of within subject time course reliability, connectivity and detectable connectivity were estimated by means of block boot strapping [11, 12, 29]. It is possible to construct pseudo timeseries which autocorrelation structure approaches the autocorrelation structure of the timeseries from which they were sampled. For this purpose, blocks with starting points S of block length b are sampled from the time courses of interest. Within his context the starting points S are drawn from an uniform distribution while the parameter b of variable length is drawn from a geometric distribution [12]. It is possible to estimate the length of b automatically using spectral estimation via the flat-top lag-windows method [11]. The flat-top lag-windows method involves choosing window sizes that are large enough to capture the autocorrelation structure of the time series, but small enough to avoid averaging over too many observations. We incorporated code provided by MFE toolbox into our pipeline [30]. Since this analysis is correlational in nature, we automatically estimated b for the seed and the target region and took the larger b value as the b value of interest that was than applied to both time courses to maintain consistency. A similar procedure was followed

for time course reliability in which the time courses of the test and retest run were considered. We also applied block bootstrapping algorithm to the time courses when residual autocorrelations were removed by means of an AR(1) model. This is allowed since the parameter b is automatically derived thus delivering a correct solution for the AR(1) corrected time courses that in ultima ratio may result in blocks of size n = 1.

**Bootstrapping at the sample level.** We aimed to investigate the impact of sample size on the within sample reproducibility of fMRI connectivity for time courses with and without AR (1) corrections. The connectome was based on the 34 coordinates of an fMRI meta-analysis [20]. To explore the relationship between sample size and reproducibility, we generated 10,000 subsamples from n = 10 up to n = 50, resulting in a total of 40 * 10,000 = 400,000 sub samples. For each sub sample, we utilized dice overlap and conjunction analysis as measures of within group reproducibility. These statistics provide insights into the level of agreement among two measurements within each individual sample. Next, we estimated the standard deviations of these reproducibility statistics across the 10,000 subsamples for each sample size. The standard deviation serves as an indicator of the between sample variability in within sample reproducibility at a given sample size n. Additionally, we computed mean within subject time course reliability, mean connectivity, and mean detectable connectivity from the connectomes from every single subsample. This results in 10,000 mean reliability and mean connectivity estimates for each sample size n. In cases where reliability is negative, we set test-retest reliability to zero. Finally, we calculated the grand mean and associated standard deviations from the 10,000 mean connectivity and mean reliability values per sample size. These values summarize the overall within sample reproducibility and connectivity for a given sample size and provide further insights into the variability observed across different samples of a particular sample size.

**Bootstrapping at the connectome level.** We constructed a connectome based on 34 coordinates deemed crucial for working memory [20]. The question arises as to whether this hypothesis-driven connectome exhibits higher reproducibility compared to randomly generated connectomes of equivalent size. To address this query, a common anatomical mask was created in 2D mesh space, representing brain areas where time courses were present across all individuals. We randomly selected 34 target vertices of interest and drew an 8mm diameter circle around each on the 2D mesh representation of the brain. We constructed a connectome featuring 561 paths, and then estimated relevant statistics, encompassing conjunction, dice overlap, and connectivity for AR(1) (un)corrected time courses. A total of 10,000 random connectomes were generated, producing 10,000 conjunctions, 10,000 dice overlap estimations, and 10,000 mean connectivity estimations, from which we derived 2,5 and 97.5 percentile intervals and standard deviations to characterize the variability in our findings.

**Thresholding policy.** It is common practice to threshold fMRI images at a Bonferroni corrected p value. As all p values obtained were far beyond Bonferroni thresholds, we decided to perform a more conservative policy and threshold on effect sizes. We utilized Cohen's effect size guidelines, with r = 0.3 denoting a medium effect and r = 0.5 denoting a large effect

## Results

### Reliability of behavioral responses and fMRI signals

In Fig 1A and 1B aspects of response time reliability are depicted while in Fig 1C the relation between response time reliability and within subject fMRI time course reliability is given. Further details about the within subject fMRI time course reliability are given in Fig 1D–1F. In Fig 1A, we compare the average response times per task item for the test and retest runs. The plots show that the first two responses of the test run were slower than the retest run, possibly due to the novelty of the situation. However, this did not seem to affect the fMRI signal, as the

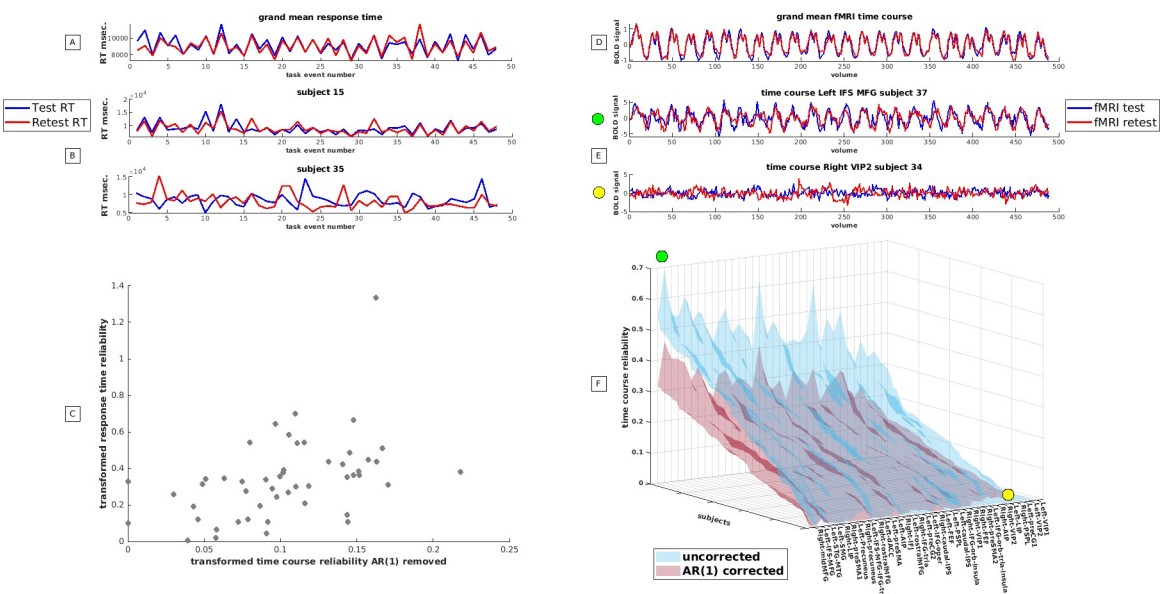

**Fig 1. Reproducibility of response time and brain measures.** A: the average response times per task event is given for a test (blue) and retest (red) run. The vertical axis reports the mean response time the horizontal axis reports the actual task event in chronological order. Mark that the response times related to the distracting Stroop task—reported on the uneven numbers—are generally shorter than the response times related to the working memory task reported on the even numbers. B: illustrates the exemplary response time curves of two individuals (blue = test; red is retest). The top curve reports the individual with the highest response reliability while the bottom reports an individual with the poorest response reliability. C: depicts the relation between within-subject time course reliability and response time reliability. Note that correlations underwent forward z-transformation to correct for nonlinear bias. Negative test-retest reliability was set at zero. D: depicts the grand mean time courses which were estimated from 1700 time courses of a test (blue) and retest run (red). E: shows fMRI time courses of two individuals that exhibited the poorest and best within subject time course reliability. The green and yellow spots indicate where the time courses depicted in E can be found in the multi-dimensional reproducibility space depicted in F. F: reports the within-subject time course reliability before (blue) and after corrections for residual auto correlations (red). We depict the 50 sorted test-retest reliabilities that could be obtained from the 50 individuals under study per brain region. Brain regions with higher average test-retest reliabilities are reported on the left while regions with the poorer average test-retest reliability are reported on the right. The vertical axis reports the height of the within-subject time course reliability and is expressed as a correlation. Note that negative correlations were set at zero.

grand mean signals were almost identical (Fig 1D). A paired t-test estimated between the mean response times of the test and retest run did not show a significant difference (p = 0.34), supporting the notion that learning is unlikely. Notably, the test-retest reliability of the average response time curves, was high (r = 0.83). The latter was not the case for the within subject response time reliability which was modest on average (mean r = 0.32, std = 0.22) although large individual differences in response behavior reliability were observed (range = 0.01–0.87). We have illustrated the latter in Fig 1B which reports the response time curves for the most reliable and unreliable individuals. Just like grand mean behavioral response curves, grand mean fMRI time courses were highly similar for test and retest runs (Fig 1D). However, this was not the case for single fMRI time courses of single individuals. We estimated the grand mean within-subject time course reliability when negative reliability was included in the estimate. For this analysis the lower and upper bound CIs were averaged separately. The grand mean within subject time course reproducibility equaled r = 0.19 (average CI 0.07–0.29) and r = 0.10 (average CI 0.01–0.19) for time courses that were subjected to AR(1) corrections (S1 Table in S1 File). On average 6.24% of the time courses exhibited negative reliability. Excluding these negative estimates from the mean lead to a grand mean within subject time course reproducibility r = 0.20 (average CI 0.09–0.31). On average 9.41% of the time courses exhibited corrupt reliability when residual autocorrelations were removed while a grand mean within-

subject time course reliability of r = 0.12 (average CI 0.03–0.20) was observed (S1 Table in S1 File). Large individual differences in within-subject time course reliability were observed for (un)corrected time course reliabilities (Fig 1E and 1F). The highest within-subject time course reliability yielded a correlation of r = 0.69 (Fig 1E and 1F) for uncorrected time courses. The depicted time courses reveal 24 Blood Oxygen Level Dependent (BOLD) response cycles that are clearly related to the 24 task cycles of the cognitive experiment (Fig 1E top).

By contrast, time courses of another individual in another region exhibit random BOLD expression despite near perfect task performance (Fig 1E bottom). Fig 1F suggests that individuals have a greater impact on test-retest reliability of the fMRI signal compared to brain regions. However, the effects of brain regions cannot be ignored. Fig 1F reveals that the right middle frontal gyrus showed the highest reliabilities on average (r = 0.31, r = 0.17 AR (1) removed), while the left ventral intra parietal sulcus exhibited the lowest reliabilities on average (r = 0.09, r = 0.05 AR (1) removed). More information about the within subject time course reliability of other brain regions can be found in the S2 Table in S1 File. Finally, we accessed whether a correlation between within-subject time course reliability and response time reliability exists. We observed a medium size correlation of r = 0.46 (r = 0.45 after AR(1) correction). We depicted the scatter plot of the relation corrected for residual auto correlations in Fig 1C while the relation without residual correction was depicted in S1 Fig in S1 File. Notably, we observed that conventional test-retest reliability (denoised r = 0.73; denoised AR(1) corrected r = 0.64) estimated at the group level is substantially higher than within-subject time course reliability.

## Effect of within-subject time course reliability on detectable connectivity

Grand mean absolute connectivity and detectable connectivity estimates and their CI are reported in Table 1. In addition, standard deviations and percentiles of the distributions are given. Negative connectivity occurred rarely meaning that in practice no difference between

**Table 1. Connectivity statistics.**

|  | mean connectivity | lower bound connectivity | upper bound connectivity | distribution percentile 2.5% | distribution percentile 97.5% | STD of distribution | percentage of paths with negative connectivity |
|---|---|---|---|---|---|---|---|
| **Connectivity Test** | 0.41 | 0.31 | 0.50 | 0.05 | 0.72 | 0.22 | 0.31 |
| **Connectivity Retest** | 0.40 | 0.29 | 0.49 | 0.03 | 0.72 | 0.22 | 0.26 |
| **Connectivity AR(1) Test** | 0.33 | 0.24 | 0.41 | 0.03 | 0.65 | 0.19 | 0.21 |
| **Connectivity AR(1) Retest** | 0.32 | 0.24 | 0.41 | 0.03 | 0.64 | 0.19 | 0.22 |
| **Detectable Connectivity Test** | 0.16 | 0.08 | 0.26 | 0.00 | 0.40 | 0.12 | 0.34 |
| **Detectable Connectivity Retest** | 0.16 | 0.08 | 0.26 | 0.00 | 0.40 | 0.12 | 0.29 |
| **Detectable Connectivity AR(1) Test** | 0.09 | 0.03 | 0.16 | 0.00 | 0.23 | 0.07 | 0.22 |
| **Detectable Connectivity AR(1) Retest** | 0.09 | 0.03 | 0.16 | 0.00 | 0.24 | 0.07 | 0.22 |

Mean connectivity and the corresponding mean confidence intervals that were estimated on the within subject single path level. We give the 2.5% and 97.5% percentiles of the distribution and the standard deviation of the distribution. The distribution consisted of 561paths*50 individuals. In addition, the percentage of paths with negative connectivity is reported

mean connectivity and mean absolute connectivity could be observed for the full sample. More details are reported in S3 Table in S1 File. Notably, CI of detectable connectivity does not cross the CI of conventional connectivity. Corrections for serial autocorrelations reduce connectivity less compared to corrections for within-subject time course reliability as the connectivity dropped from r = ~0.4 to r = ~0.32 in the case of autocorrelation corrections, while it dropped from r = ~0.4 to 0.16 in the case of reliability corrections. Autocorrelation corrections had a similar effect when applied to mainstream connectivity or to detectable connectivity, as the absolute difference between uncorrected versus corrected data was roughly between r = 0.08 and r = 0.07 respectively. Such differences are expected given the high degree of autocorrelations reported in S1 Text in S1 File. However, the relative effect of the autocorrelation correction was much stronger in the case of detectable connectivity, as in this case the amount of connectivity was almost halved due to the lower initial connectivity level.

We visualized how connectivity collapses when autocorrelation corrections and reliability corrections are performed in Fig 2. In this figure, the whole range of connectivity is depicted per individual for all participants. In uncorrected data, a large spread of connectivity values exists within individuals that roughly range from r = 0.8 to r = -0.2. The correction for autocorrelations changed the rank of the correlations within the individual, which can be viewed by the occurrence of spikes on the mesh; however, the connectivity range was well maintained at 0.8 to -0.2. Corrections for within-subject time course reliability had a strong effect on the connectivity range since it ranged from 0.4 to roughly zero in the best case. The few negative connectivities that were previously detected vanished as they did not appear to be reliable. The ranking of the connectivity strength within an individual fell into disarray, with many values that were once high in the uncorrected case vanishing into obscurity. This was even more the

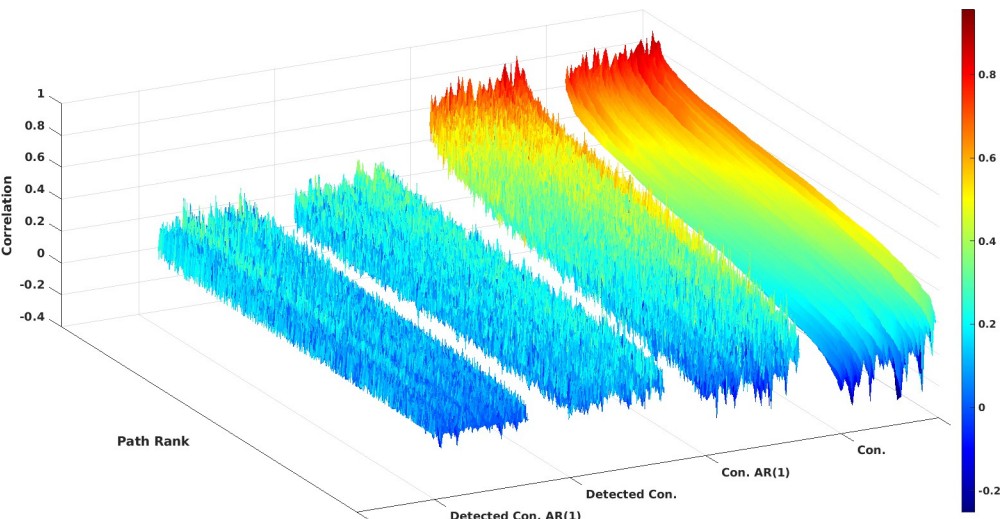

**Fig 2. Impact of corrections on connectivity.** This figure illustrates the impact of corrections for auto correlations and within-subject time course reliability on connectivity estimates within the connectome of an individual. Four surfaces display the results for uncorrected connectivity (Con), connectivity corrected for auto correlations (Con AR (1)), connectivity corrected for within-subject time course reliability (Detected Con), and connectivity corrected for both within-subject time course reliability and auto correlations (Detected Con. AR(1)). The connectivities of the test and retest run were averaged. Subsequently the connectivity correlations of uncorrected connectivity (Con) were sorted per individual along the X axis (path rank) where individuals with low mean connectivity were found on the left of the surface and individuals with high mean connectivity on the right. The rank of the original uncorrected connectivity correlation was maintained when corrections were performed. Note how the order of the correlations within the individual connectomes fall into disarray when corrections are performed.

case when additional corrections for autocorrelations were applied to the individual detectable connectivity maps, as the surface was changed into an unorganized "forest" of spikes.

In a next step we compared the connectivities of the test and retest run and took the smaller correlation as a measure of conjunction. Formal tests between the conjuncted correlations of the four ways to estimate connectivity were statistically significant attesting that corrections for residual autocorrelations and within-subject time course reliability have a very dramatic effect on the height of the achievable connectivity that in one case was not estimable as the p values approached zero (S2 Fig with its attached table in S1 File).

### Effect of within-subject time course reliability on NHST based group reproducibility

For the NHST analysis, we did not consider the absolute but the observed connectivity estimates since this reflects current practice. The results of the conjunction analysis collapsed over the test and retest run resulted in a median t-statistic of 15.65 (p = $1.07e^{-20}$) for the conventional connectivity analysis and 10.17 (p = $1.15e^{-13}$) when detectable connectivity was the object of analysis. These statistics diminished to 14.71 (p = $1.32e^{-19}$) and 9.12 (p = $3.91e^{-12}$) for the connectivity and detectable connectivity estimates respectively, when residual autocorrelations were removed from the time series (Fig 3A). The scatter plots which depict the relation among the distinct ways to estimate connectivity reveal that the observed diminishment in t statistics—that resulted from the correction methods—is an orderly process (S3 Fig in S1 File).

The number of paths detected and the value of the dice overlap remained maximal between roughly p = 0.74 and p = $2.54e^{-6}$ irrespective of the connectivity measure under study (Fig 3B). However, this does not imply that corrections for within-subject time course reliability and

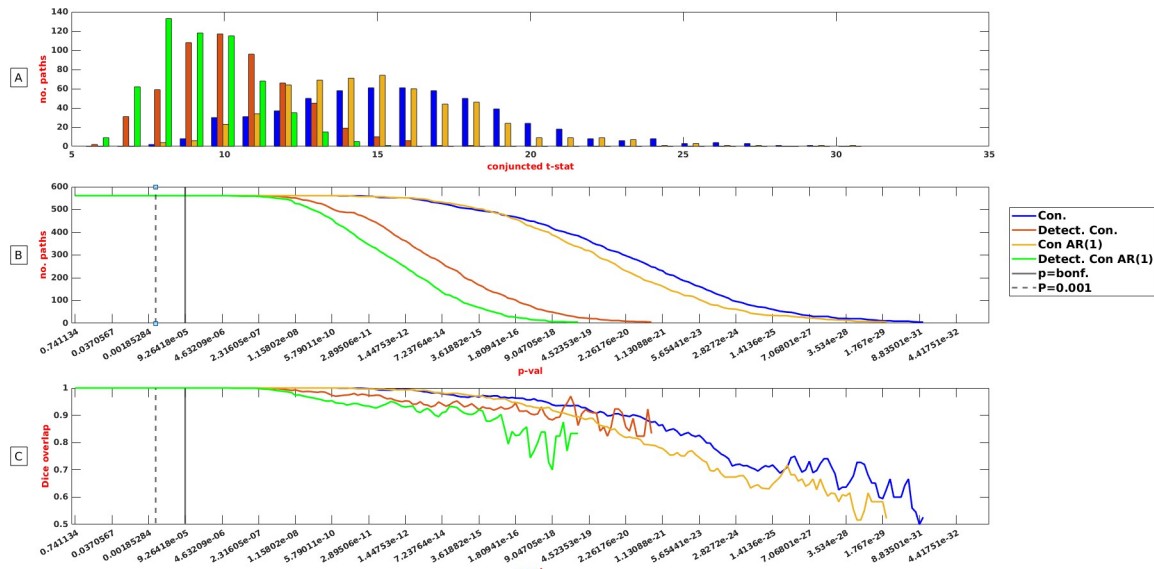

**Fig 3. Thresholds and group reproducibility.** Connectivity (blue), connectivity AR (1) removed (yellow), detectable connectivity (red) and detectable connectivity AR (1) removed (green) response to conjunction and dice analysis at a particular significance threshold in a sample of 50 subjects. A: 561 paths of a test and retest run were subjected to NHST. Subsequently the smaller t value of the test and retest run was taken and depicted for the four kinds of connectivity measures separately. The vertical axis represents the number of paths that were commonly detected in two independent observations at a particular minimal t-statistics value.B: 561 paths of a test and retest run were subjected to NHST. The vertical axis reports the number of paths that survived conjunction analysis at a particular significance threshold. The interrupted vertical line represents the commonly used p = 0.001 threshold while the solid line represents a Bonferroni corrected threshold. C: 561 paths of a test and retest run were subjected to NHST. The vertical axis represents the dice overlap coefficient that could be detected at a particular significance threshold.

residual autocorrelations had no effect on the number of detected paths at the group level. The number of paths detected at half maximum (561/2 = ~280 paths) was associated with a p value of $9.21e^{-21}$ for conventional connectivity while it dropped to a p value of $3.56e^{-12}$ when within subject time course reproducibilty and residual auto correlations were considered (Fig 3B). The Dice overlap remained high far beyond Bonferroni corrected p-values (Fig 3C). Since Bonferroni corrected (p = 0.05/561) connectivity maps were identical for all reproducibility schemes (561 paths out of 561 paths), we opted for an extremely conservative threshold of $1.69e^{-16}$, which refers to a correction for $2.95e^{-14}$ comparisons (S4 Fig) S1 File.

As an alternative approach we utilized Cohen's effect size guidelines, with r = 0.3 denoting a medium effect and r = 0.5 denoting a large effect [31]. We averaged the positive and negative AR(1) corrected connectivity correlations in a path-wise manner across 50 individuals and applied thresholds of r>0.3 (Fig 4A and 4B) and r>0.5 (Fig 4E and 4F) to the resulting images. Since detectable connectivity maps did not reach the correlation criterion of r >0.3 different thresholds were established to compare the distinct maps. We determined the correlation threshold at which the conjuncted detectable connectivity maps exhibited the highest dice overlap with the conventional univariate maps thresholded at r>0.3 or r>0.5 (Fig 4D). Further analysis revealed that detectable connectivity maps show good resemblance with conventional connectivity maps (Fig 4C) when liberal thresholds (r>0.3) where applied while this was not the case (Fig 4G) for conservative thresholds (r>0.5). We subjected the conventional connectivity maps to conjunction analysis and took the smaller correlation of two sessions to obtain maps that portray reproducible result. We present the top 5% conjuncted correlations of conventional and detectable connectivity maps corrected for residual correlations in Fig 4H. The two distinctive maps portray contrasting images of the brain, with only minimal similarities depicted in bright yellow. The conjuncted conventional maps (depicted in orange) suggests that the highest intra hemispheric connectivity density takes place in the right hemisphere. Furthermore, this analysis points to the existence of dorsolateral pre frontal hubs that mainly connect to the contralateral frontal and contralateral parietal systems. Conversely, the detectable connectivity maps (depicted in red) show highest interhemispheric connectivity density in the left hemisphere and points to the existence of two inferior parietotemporal hubs that primarily connect with the contralateral/ipsilateral parietal and contralateral frontal system.

## Effects of sample size on group reproducibility

To thoroughly assess the influence of sample size on within sample reproducibility, a comprehensive bootstrap analysis was conducted, encompassing the entire sample size range from 10 to 50 individuals. We depicted the results of this analysis for a broad range of significance thresholds in S5, S6 Figs in S1 File. Here we restrict ourselves to results that were obtained at a Bonferroni corrected threshold (p = 0.05/561 paths). Result of the simulation suggest that the number of paths that survive conjunction analysis is very sensitive to samples size while this is less the case for dice overlap measures and connectivity/reliability point estimates. In Fig 5A we observe a significant disparity between the number of paths detected using conventional connectivity measures compared to detectable connectivity measures. For sample sizes of n = 10, 293 paths were detected with conventional measures, while only 36 paths were discovered when connectivity was corrected for within subject time course reliability. However, as sample size approached 25 individuals, this "conjunction gap" disappeared. Fig 5B demonstrates that the variability in the number of detected paths diminished when a sample size of 30 was reached. This suggests that there were no discernible differences in within sample reproducibility among samples of n = 30. While the detected number of paths responded to the sample size and corrections for within subject time course reliability, this was less the case for the

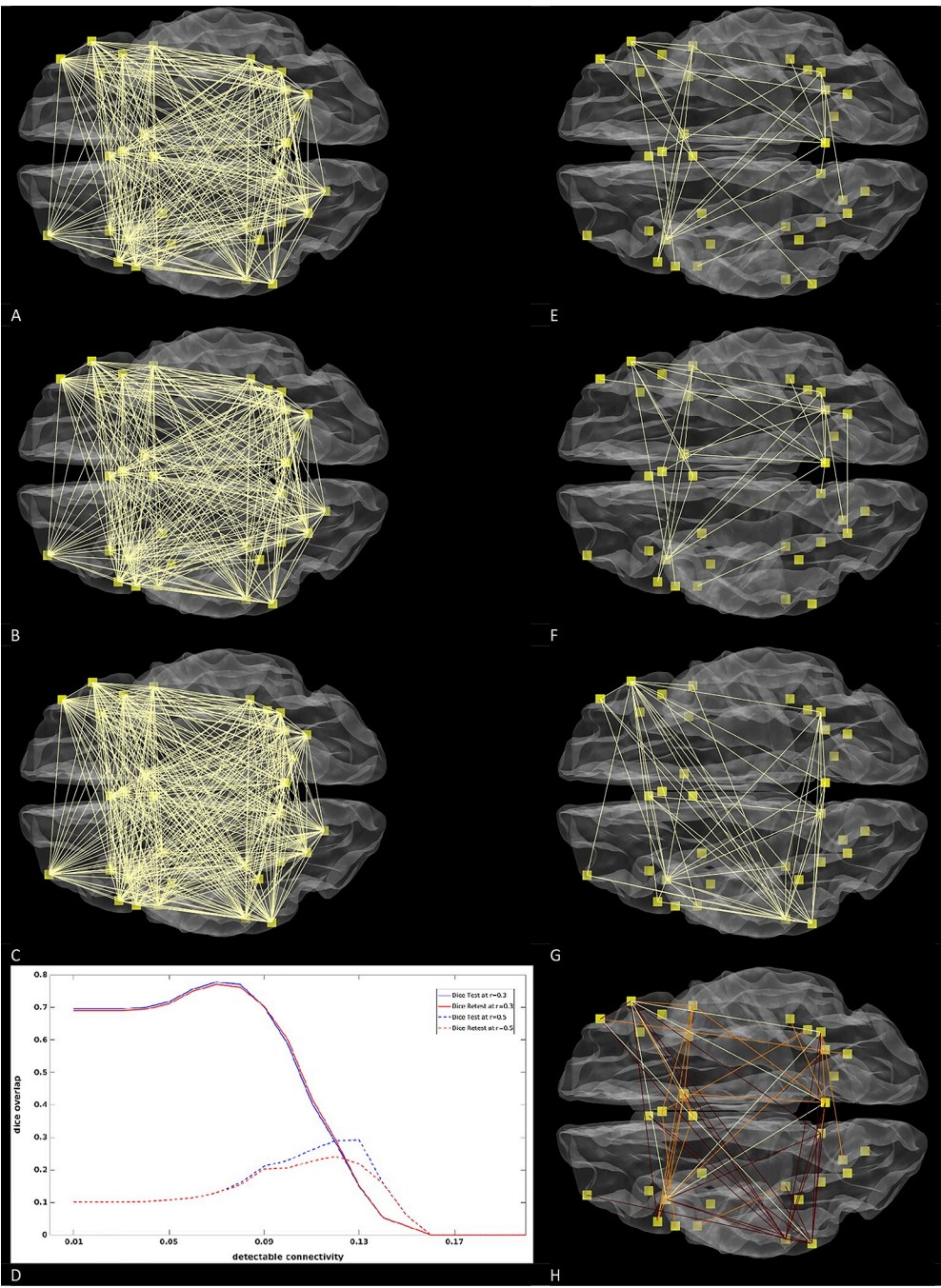

**Fig 4. Effect of corrections on connectivity maps.** A: Conventional connectivity obtained from the test run r>0.3. B: Conventional connectivity obtained from the retest run r>0.3. C: Depicts the conjuncted detectable connectivity map that exhibited the highest possible likeness (dice~0.77) with conventional connectivity maps set at r>0.3. The threshold of this map (r>0.07) was found by averaging the dice overlap curves of the test and retest run reported in D: We estimated at which detectable connectivity threshold the highest dice overlap between detectable connectivity and conventional univariate connectivity maps could be obtained when conventional connectivity was either set at r>0.3 (solid) or r>0.5 (dotted lines). The vertical axis represents the dice overlap between univariate connectivity and detectable connectivity while the horizontal axis represents the strength of the detectable connectivity. Note that the detectable connectivity was obtained after conjunction analysis while this was not the case for conventional connectivity. E: Conventional connectivity obtained from the test run r>0.5. F: Conventional connectivity obtained from the retest run r>0.5. G: Depicts the conjuncted detectable connectivity map that exhibits the highest possible likeness (dice~0.27) to the conventional connectivity maps set at r>0.5. The threshold of this map (r>0.12) was found by averaging the dice overlap curves of the test and retest run reported in D. H: The smallest correlation of a test and

retest run was taken as a measure of conjunction. Subsequently the top 5% correlations of conventional and detectable connectivity were depicted. Yellow refers to what both networks have in common whereas orange refers to reproducible results that are typical for conventional connectivity while dark red refers to results that are typical for detectable connectivity maps.

dice overlap measures that roughly varied between 0.76 for n = 10 to 1.0 for n = 30 (Fig 5C). When sample sizes were small, there was considerable variability in "within sample overlap" between the selected samples that was in particularly visible when connectomes were built from detectable connectivity measures (Fig 5D).

By contrast point estimates of the four connectivity measures under study expressed as correlations were stable (Fig 5E) and exhibited only small standard deviations irrespective (Fig 5F) of the sample size while a similar picture was observed for within-subject time course reliability (Fig 5G and 5H). Similar to our observations for the full sample of n = 50, corrections for residual correlations and within-subject time course reliability had a very severe impact on the detectable amount of connectivity. Finally, the use of excessively conservative p values is not recommended as it negatively impacts both the reproducibility within a sample, as measured by the Dice overlap, and the variability in within sample reproducibility. This holds true regardless of the size of the sample being studied (S6 Fig in S1 File).

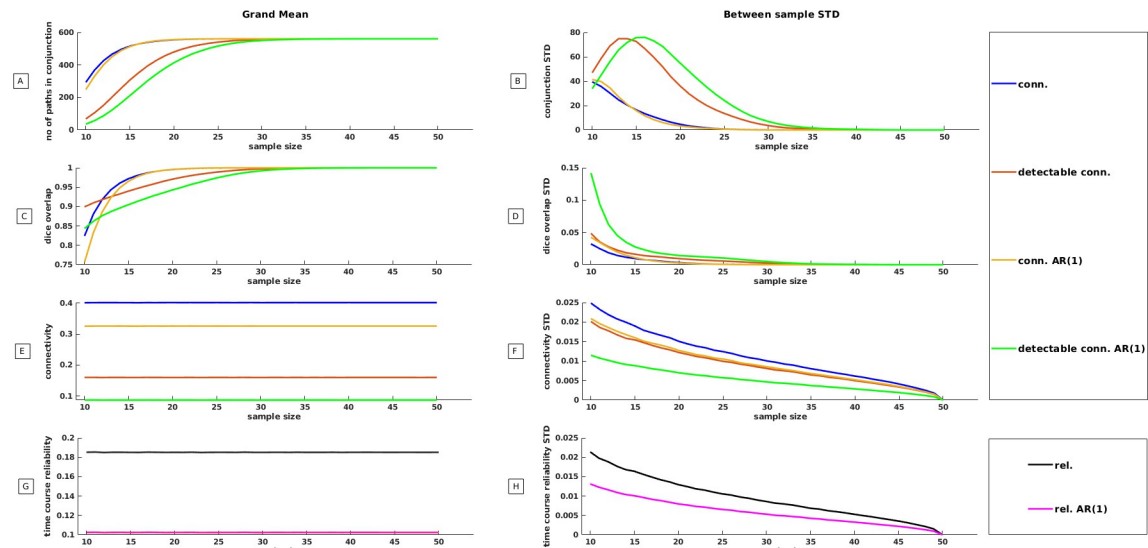

**Fig 5. Effect of sample size on reproducibility.** Outcomes of a Monte Carlo simulation designed to estimate the influence of sample size on various aspects of functional connectivity reproducibility. Throughout each individual Monte Carlo iteration, a random sample was generated, ranging from n = 10 to n = 50 subjects. A comprehensive total of 10,000 Monte Carlo iterations were executed per sample size. Essential statistics were derived from both test and retest sessions for each iteration, resulting in 10,000 observations for each statistic per sample size. On the left side, we depict the grand mean reproducibility and connectivity statistics, while the right side illustrates the standard deviation of these statistics. The latter provides insight into the extent to which within sample reproducibility and connectivity varies between samples for a particular sample size. A: In this panel, NHST was applied to 561 paths from test and retest runs within each iteration. The larger p-value between the test and retest runs was taken as a measure of conjunction. The vertical axis represents the mean count of paths that exhibited consistent detection across two independent observations, at a Bonferroni-corrected p-value (conjunction analysis) for a specific sample size. B: Variability in the number of detected paths among different samples. C: 561 paths from test and retest runs underwent NHST per iteration. The vertical axis corresponds to the grand mean dice overlap coefficient attainable at a Bonferroni-corrected p-value for a specific sample size. D: Variability in dice overlap among different samples for a given sample size. E: Grand mean connectivity was computed for each type of connectivity per sample size. F: Variability in mean connectivity across samples for a given sample size. G: Grand mean within-subject time course reliability estimated per sample size. H: Variability in mean within-subject time course reliability across samples.

## Effect of connectome selection on group reproducibility

The effects of hypothetical versus random connectomes were in particularly visible for the number of paths that survived conjunction analysis (Fig 6A) and the height of the connectivity strength that could be achieved (Fig 6C) while it was less pronounced for the within group reproducibility as captured by the Dice overlap measure (Fig 6B). While the number of paths in conjunction was maximal irrespective of the connectivity measure under study in the hypothetical connectome, this was not the case for random connectomes. We visualized the effects of the randomization procedure on the conjunction analysis in Fig 6A and report the exact numbers in S4 Table. The difference between hypothetical (561 paths) and random connectomes (319 paths) was less pronounced for connectivity in the usual sense of the word but highly visible when the test-retest reliability of the time courses was taken into account (detectable connectivity 190 paths). This figure collapsed to 131 paths when corrections for residual auto correlations were performed. The results of the study showed that the randomization of connectome coordinates and the choice of the connectivity measure had a significant impact on the number of detected paths. Fig 6B demonstrates that within sample reproducibility, as measured by Dice overlap estimates, remained relatively high, ranging from 0.75 to 1, regardless of the number of paths used for estimation. This suggests that the group reproducibility of connectomes remained high even when connectomes were randomly selected and corrected for within subject time course reliability. It should nonetheless be stated that the observed reproducibility of the hypothetical connectome was comfortably beyond the realm of the random connectome upper bound statistics. Fig 6C illustrates that the height of the detectable connectivity was significantly affected by connectome randomization. In fact, the maximum achievable height of connectivity in random connectomes was only about half of what could

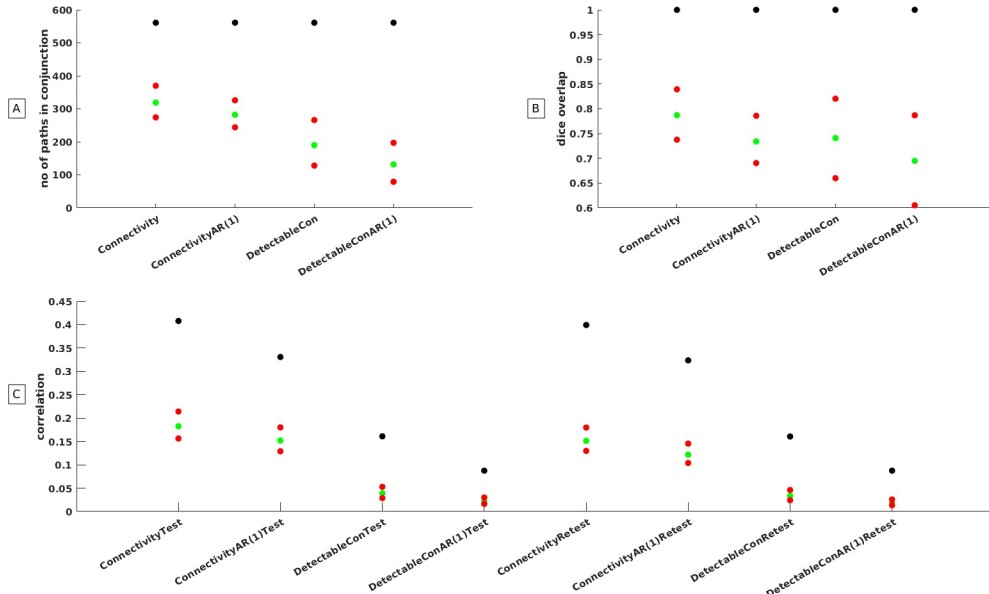

**Fig 6. Hypothetical versus random connectomes.** Statistics from the hypothetical connectome derived from meta-analysis (depicted as black dots) in relationship to the 2.5 and 97.5 percentiles of statistics that were obtained from 10,000 random connectomes (depicted as red dots). Green dots refer to the grand mean of the 10,000 random connectomes. Panel A: Visualizes the number of paths that survived conjunction analysis at a Bonferroni corrected p value for the four distinct ways to estimate connectivity. Panel B: Visualizes the Dice Overlap that could be obtained at a Bonferroni corrected p value for the four distinct ways to estimate connectivity. Panel C: Visualizes the mean connectivity strength that could be obtained for the four distinct ways to estimate connectivity.

be achieved in hypothetical connectomes. The detectable connectivity approached zero when connectomes were randomly selected. These findings highlight the critical role of connectome coordinates and within-subject time course reliability in determining the number of detected paths and the strength of connectivity.

## Discussion

*Our analysis revealed that within-subject time course reliability has a profound effect on the amount of connectivity that can be detected.* A conventional analysis of our data revealed that the average connectivity of the connectome may amount to approximately 0.4. This figure might be overestimated as it dropped to an average connectivity of 0.16 when within-subject time course reliability was taken into account. When corrected for residual auto correlations, the detectable amount of connectivity decreased even further to 0.09. Taking the square root of 0.09, we can infer that only 0.081 percent of the observed connectivity might be attributed to cognitive-related signal fluctuations. The drastic reduction of within-subject time course reliability after AR(1) corrections needs to be replicated in future studies. We observed that very large individual differences in within subject response time reliability exist (range = 0.01–0.87). These individual differences in response time reliability were correlated (r~0.45) with individual differences in within subject time course reliability. This finding corroborates previous observations that cognitive aspects of tasks are related to within-subject time course reliability [1]. Notably, we observed that conventional test-retest reliability (denoised r = 0.73; denoised AR(1) corrected r = 0.64) estimated at the group level is substantially higher than within-subject time course reliability [1].

*Our analysis revealed that sample size and within-subject time course reliability have a profound effect on the number of paths that can be reproduced via NHST based group studies.* For a sample size of 10 subjects, 293 paths were detected at a Bonferroni corrected p value, while only 36 paths were detected when connectivity was corrected for within-subject time course reliability and residual autocorrelations. The gap in the number of paths detected by the various connectivity measures disappeared when sample sizes approached 30. The reproducibility of NHST based group effects in larger samples (n = 50) was greatly affected by within subject time course reliability. The median p-value at which paths were detected changed from $1.07e^{-20}$ for conventional connectivity measures to $3.91e^{-12}$ when connectivity was corrected for within-subject time course reliability and residual auto correlations. However, these effects were found beyond Bonferroni corrected p values and had no impact on Bonferroni corrected maps. Our study findings indicate that group studies characterized by high reproducibility in terms of NHST could benefit from additional information regarding the detectable connectivity upper limit. This is crucial for a correct interpretation of the data because detectable connectivity may be very low despite high group reproducibility.

*Our Monte Carlo analysis revealed that detectable connectivity and group reproducibility is affected by the exact coordinates of the nodes that are used to construct the connectome.* The simulations demonstrate that the hypothetical connectomes derived from a working memory meta-analysis is significantly more reliable when compared to the 10,000 randomly generated connectomes. As an example, we want to mention that the average connectivity of the hypothetical connectome was 0.4 while it was only 0.18 (test) and 0.15 (retest) for random connectomes. The detectable connectivity for the hypothetical connectome was r = 0.09 when time courses were corrected for within subject time course reliability and auto correlations while it was r = 0.02 when connectomes were randomly selected. The latter difference does not seem to be impressive but it has nonetheless consequences for the group reproducibility of the images on the basis of conjunction analysis. While all 561 paths of the hypothetical

connectome were reproduced at a Bonferroni corrected p value after rigorous corrections, only 131 paths were reproduced when connectomes were randomly selected. This demonstrates that there is some merit in the use of meta-analysis results when constructing hypothetical connectomes. The effectives of the hypothetical connectome were in particular visible in S4 Fig in S1 File were a p $<1.69e^{-16}$ was needed to differentiate group statistic maps from each other. We conclude that hypothetical connectomes that are derived from brain activation studies are also relevant for connectivity studies.

*What can be done to enhance within-subject time course reliability*? A promising approach to improve within-subject time course reliability is to simply average task-related time courses obtained from multiple independent fMRI sessions. Alternatively, one might incorporate several time courses of the same individual that measure the same task into a multivariate statistical model such as structural equation modeling. Such multivariate techniques have been shown to improve reliability on the group level [5]. A cheaper alternative might be to refine high and low-pass filters used to separate noise from signal. Current fMRI software packages typically employ a one-size-fits-all filter approach, which may not be optimal. Theoretically, designing filters based on the expected shape of the task-related response could effectively separate noise from signal without excessively altering the autocorrelational structure of the time courses. Possibly ultra-high field scanners with more favorable signal to noise ratios could detect higher levels of within-subject time course reliability. Within this context, it would be beneficial to assess the contributions of signal-noise ratio under different measurement conditions and post-processing techniques through a sensitivity analysis.

*How do our results generalize to other studies and populations*? Our study demonstrated complete group connectome reproducibility at P $< 2.54e^{-6}$ even when stringent corrections were applied. The nodes utilized in this investigation were originally derived from a working memory study, suggesting that findings from brain activation studies can be generalized to connectivity studies across distinct samples. The highest mean within-subject time course reliability of a specific region was slightly higher in this study (r = 0.31) compared to the study of our colleagues (r = 0.25) [1]. Whether this is a fair comparison is debatable since field strength and task designs were different. The findings from our group reliability analysis suggest good (0.64 to 0.73) reliability which surpass expectations based on recent fMRI meta-analyses which estimate that group reproducibility of fMRI does usually not exceed r = 0.4 [3, 4]. As the resting state time courses obtained from a test and rest run are inherently not synchronized due to the wandering nature of the mind, it is impossible to make any meaningful statements about the detectable connectivity of resting state studies. This draws a cast on the clinical applicability of this attractive technique in the context of pre surgical mapping and other critical applications of fMRI. It is obvious that our data are limited to healthy individuals so that we cannot draw any conclusions about clinical populations that may exhibit different within subject time course reliability. In our view, it is important to examine the within-subject time course reliability of language and hand functions using robust block designs, as these functions are essential for effective pre-surgical mapping.

*How can we interpret conventional NHST-based reproducibility measures and detectable connectivity measures from a neural perspective*? Conventional conjunction-based reproducibility measures principally detect commonality in brain functional anatomy. However, neural communality does not imply communality in cognitive processes. By contrast detectable connectivity measures, capture commonality in both brain anatomy and the temporal aspects of neural information flow [30]. Within subject time course reproducibility may enhance our understanding of brain connectivity because detectable connectivity maxima differ from conventional connectivity maxima. Within this context, cognitively relevant paths are not

necessarily the ones with the highest correlations because detectable connectivity is low compared to conventional connectivity.

## Limitations

Some limitations of our work should be discussed. First, the slow event related design used in this study exhibited relatively rapid signal shifts within a trial. Our study may not generalize to conventional block designs that are more robust and therefore possibly more reliable.

Another limitation is that the sub-samples of the Monte Carlo simulation were drawn from a data set which consisted of 50 individuals only. While there might be sufficient between-sample variability in smaller samples (n = 10) this is maybe less the case for larger samples (n = 50). The curves obtained should be interpreted with care and partly be seen as a proof of the principle.

The Monte Carlo simulation clearly showed that a connectome that was based on hypothetical brain activation maxima derived from meta-analysis yields far more reproducible results compared to connectomes that have been selected randomly. However, this does not imply that meta-analysis derived connectomes are the best possible solution. As an alternative one could derive connectomes from hypothesis free methods such as independent component analysis, clustering and deep factor learning [32–34]. It was beyond the reach of this study to prove that connectomes derived from resting state and structural data by means of hypothesis free methods outperform brain regions that have been based on the neural correlates of working memory.

While pre whitening by means of an AR(1) model is a standard procedure that is performed by most brain imaging software our analysis showed that very small auto correlations (r<0.05) remain at lag 1 to lag 3. It is not impossible that the reported within-subject timcourses reliability is still slightly overestimated given that very small correlations were still present at the reported lags. We feel that the bias introduced by these minute autocorrelations is minimal and in practice may not be not tracible.

As a further limitation one might argue that the concept of test-retest reliability requires that the repeated measures are obtained from an identical set of individuals at a specific scanner. It is possible that the observed reproducibility is limited by the measurement accuracy of this specific machine. It remains elusive whether the results obtained in this project can be generalized to other machines as well.

## Conclusion

Overall, it would be beneficial for distinct scanner centers to publish the within-subject time course reliability of their scanners with standard protocols. The many fMRI connectivity studies published to-date may be based on inflated estimates and adding this additional dimension of reproducibility may help reach a more nuanced interpretation and assessment of the relevance of this large literature.

## Supporting information

**S1 File.**
(DOCX)

**S1 Data.**
(ZIP)

## Acknowledgments

We thank Hanna Weber for her insights around bootstrapping.

## Author Contributions

**Conceptualization:** Jan Willem Koten.

**Data curation:** Jan Willem Koten.

**Formal analysis:** Jan Willem Koten, Hans Manner, Cyril Pernet, John P. A. Ioannidis.

**Funding acquisition:** Guilherme Wood.

**Investigation:** Jan Willem Koten.

**Methodology:** Jan Willem Koten, Hans Manner, Cyril Pernet, Andre Schüppen, John P. A. Ioannidis.

**Project administration:** Guilherme Wood.

**Resources:** Jan Willem Koten.

**Software:** Jan Willem Koten, Andre Schüppen.

**Supervision:** Guilherme Wood.

**Validation:** Jan Willem Koten.

**Visualization:** Jan Willem Koten.

**Writing – original draft:** Jan Willem Koten, John P. A. Ioannidis.

**Writing – review & editing:** Jan Willem Koten, Dénes Szücs, John P. A. Ioannidis.

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
