## [Decision Letter · Decision Letter 0]

12 Apr 2024

PONE-D-24-04077When most fMRI connectivity cannot be detected: insights from time course reliabilityPLOS ONE

Dear Dr. Koten,

Thank you for submitting your manuscript to PLOS ONE. After careful consideration, we feel that it has merit but does not fully meet PLOS ONE’s publication criteria as it currently stands. Therefore, we invite you to submit a revised version of the manuscript that addresses the points raised during the review process.

**Two experts in the field have carefully reviewed the manuscript entitled, "When most fMRI connectivity cannot be detected: insights from time course reliability". Their comments are appended below.**

**Both referees acknowledged the manuscript is fairy well-written though this submission is not acceptable because this submission had  several serious drawbacks .**

**The first reviewer pointed out the authors should describe their own results and their corresponding insights in more detail. They need more explanation regarding the statistical methods, furthermore they should display more graphical presentation for easier reader understanding.**

**The second referee gave detailed serious concerns which arose from every aspect of the manuscript. I, this Academic Editor, agrees such critiques are sure to strengthen this study.**

**I will make the final decision after receipt of the revised manuscript and the reply message to each comment.**

We look forward to receiving your revised manuscript.

Kind regards,

Manabu Sakakibara, Ph.D.

Academic Editor

PLOS ONE

Journal Requirements:

This study was funded by the FWF grant P 22577-B18. Wood

We thank Hanna Weber for her insights around bootstrapping. The authors acknowledge the publication fees by the University of Graz. This study was funded by the FWF grant P 22577-B1

This study was funded by the FWF grant P 22577-B18. Wood

Reviewers' comments:

Reviewer's Responses to Questions

**Comments to the Author**

1. Is the manuscript technically sound, and do the data support the conclusions?

Reviewer #1: Partly

Reviewer #2: Yes

2. Has the statistical analysis been performed appropriately and rigorously? 

Reviewer #1: No

Reviewer #2: Yes

3. Have the authors made all data underlying the findings in their manuscript fully available?

Reviewer #1: No

Reviewer #2: Yes

4. Is the manuscript presented in an intelligible fashion and written in standard English?

Reviewer #1: Yes

Reviewer #2: Yes

5. Review Comments to the Author

**Reviewer #1: **This paper primarily investigates the reliability of functional magnetic resonance imaging (fMRI) and its correlation with individual time course reliability. Several assessments and critiques can be made:

Clear Objectives and Framework: The paper explicitly defines its research objectives and framework, namely studying the reliability of fMRI, detectable connectivity, and how these are influenced by individual time course reliability. The objectives are clear, and the logic is coherent.

Theoretical Foundation: The paper cites numerous studies, providing a solid theoretical foundation. Particularly, referencing Nunally's theory, it suggests that connectivity ceilings cannot exceed the individual time course reliability of two brain regions.

Research Methodology: The research methodology is well-structured, including steps such as data collection, processing, and statistical analysis. Detailed descriptions are provided, especially regarding participant selection and task design.

However, there are areas for improvement:

Presentation of Results: While the paper extensively describes the research methodology, it lacks any presentation of results. Ideally, the paper should include a results section detailing experimental findings and provide interpretation and discussion of these results.

Complex Statistical Methods: The paper employs some complex statistical methods, which may require further explanation and elucidation for better reader comprehension. Relating p-values to FDR correction to present statistical findings could enhance clarity.

Data Visualization: There is no mention of data visualization in the paper. Visualization tools can aid reader understanding of research findings. Therefore, incorporating charts or other visual elements to showcase key findings is recommended. Specifically, presenting statistical differences in brain network maps for a certain disease, such as ADHD (DOI: https://doi.org/10.1016/j.knosys.2022.109082) and PD (DOI: 10.1109/TCBB.2023.3252577), compared to a normal control group would be beneficial.

Consistent Terminology Usage: Throughout the paper, consistency in the usage of concepts and terminology should be maintained to avoid reader confusion. For instance, if "within-subject time course reliability" is used in the text, it should be consistently utilized throughout the paper without substitution.

Critical Reflection on Methods: When describing research methods, incorporating discussion on method limitations, such as potential biases in data collection or issues encountered in statistical analysis, can enhance the paper's comprehensiveness and reader understanding.

In conclusion, while the paper is well-designed overall and contributes to the fields of neuroimaging and psychology, there is room for improvement in the presentation of results and explanation of statistical methods.

**Reviewer #2:** This study investigates the influence of within-subject time course reliability on the detection of connectivity and group effects using NHST, while accounting for variables such as sample size and spatial distribution of connectome nodes. The study performs a number of detailed analyses to tease apart the impact of these factors on the reliability and interpretability of functional connectivity measures in neuroimaging research. The paper is well written and makes an important point. To enhance clarity and the study's impact, I propose significant revisions to elaborate on these aspects. I have a few suggestions that could further enhance the clarity and impact of the study.

Major:

1. The introduction section would benefit from a more explicit statement of the paper's objectives. While the aim to investigate the impact of within-subject time course reliability on connectivity detection and group effects is implied, a clearer articulation of the specific research questions or hypotheses would enhance reader understanding.

2. Additionally, the introduction occasionally suffers from a dense presentation of technical terms and concepts, which could potentially alienate readers unfamiliar with the intricacies of fMRI methodology.

3. Some of the methods suffer from slightly extreme language which is easily falsifiable, one can never ‘rule out the possibility” that “any lack” was due to poor performance or motion. Pease rework the text to be more balanced throughout.

4. The abstract mentioned 0.81 percent (is this correct, less than 1%?)…also where are the results that show this? It was not obvious from the paper.

5. Figures could be improved (e.g. figure on, needs legends for blue/reg plots for all panels)

6. Could the authors also plot the timecourse activity task effects? So not just the task-modulated connectivity, as these are measuring different things.

7. While the Discussion section effectively addresses the impact of within-subject time course reliability on connectivity measures, could the authors provide additional insights into potential strategies for mitigating the effects of variability in fMRI studies? For example, are there specific methodological approaches or analytical techniques that could help account for individual differences in time course reliability and improve the robustness of connectivity analyses?

8. Additionally, considering the limitations discussed, do the authors have any recommendations for future research aimed at addressing these challenges and further validating these findings across diverse experiments. It may be beneficial to explore the potential implications of considering whole-brain functional connectivity in addition to the specific regions investigated in this study.

9. The drastic reduction in average connectivity and decrease in detectable connectivity after correcting for residual autocorrelations needs further validation and replication of the findings to ensure the robustness and reliability of the conclusions drawn from the data. What are potential confounding factors in connectivity analyses and how corrections reflect true connectivity levels requires careful consideration.

10. Furthermore, the authors are using anatomically based/fixed ROIs and thus voxels within the ROIs may not be temporally coherent. They thus might consider at least discussing the potential utility of data-driven/probabilistic parcellation/decomposition approaches, such as independent component analysis (ICA), for estimating brain components and their connectivity, or using one of the existing FC+clustering parcellation approaches. The rationale behind connectome selection and its potential impact on connectivity measures and group-level analysis should be highlighted.

o Glasser, Matthew F., et al. "A multi-modal parcellation of human cerebral cortex." Nature 536.7615 (2016): 171-178.

o Du, Yuhui, et al. "NeuroMark: An automated and adaptive ICA based pipeline to identify reproducible fMRI markers of brain disorders." NeuroImage: Clinical 28 (2020): 102375.

11. Similarly, the incorporation of 34 MNI coordinates for reading from external data is suboptimal to using functional localization in held out data, this will reduce task effects and should be noted as a limitation. Techniques such as selecting upper quartile or other voxel subsets within ROIs can help, but are also a bit tricky to implement without inducing bias.

12. Additionally, the choice of an 8 mm radius needs further discussion. Was this a ‘smooth with’ or ‘smooth to’ 8 mm? How sensitive are the results to the smoothing kernel?

13. Averaging reliability estimates across paths may oversimplify the assessment of reliability, particularly if there is substantial variability across different paths. Individual paths may exhibit different levels of reliability and averaging them together could obscure important patterns in the data and averaging across participants may overlook this variability.

14. The appropriateness of the AR(1) model depends on the underlying structure of the data, and studies have argued that AR(1) is not a sufficient model for neuroimaging data. In addition, given the autocorrelation is strongly related to the hemodynamic response function, which can vary in interesting ways across populations, it may also be important to evaluate the autocorrelation for such effects. These issues could be discussed in the manuscript.

15. How do these findings generalize to other datasets? Robustness checks and sensitivity analyses can help assess the stability of the results across different conditions. The reproducibility of the results should be discussed in manuscript. Do other brain regions or functional networks analysis support these findings?

Minor:

1. The manuscript would benefit from a clearer separation between the methods and results sections. Specifically, it appears that certain aspects of the results, such as the content found in the last paragraph of page 13, contain information that would be more appropriately situated within the methods section.

6. PLOS authors have the option to publish the peer review history of their article (what does this mean?). If published, this will include your full peer review and any attached files.

Reviewer #1: No

Reviewer #2: No

---

## [Author Response · Author response to Decision Letter 0]

26 Jun 2024

We would like to thank the reviewers for the many useful comments most of which we have tried to implement in the manuscript. 

Presentation of Results: While the paper extensively describes the research methodology, it lacks any presentation of results. Ideally, the paper should include a results section detailing experimental findings and provide interpretation and discussion of these results.

The previous version of the manuscript consisted of sperate method and a result section. But they may have been difficult to distinguish, as the font size used for the titles was small. We have now made the fonts of the titles larger for clearer identification of where the method and result sections begin and end. The method section covers approximately 2600 words in the main text and around 2700 words in the supporting materials, while the results and its discussion section in the main text comprise around 3500 words. 

Complex Statistical Methods: The paper employs some complex statistical methods, which may require further explanation and elucidation for better reader comprehension. Relating p-values to FDR correction to present statistical findings could enhance clarity.

We agree that the explanation of the methods is very technical and may need some extension as some of the methods employed are new in the field of neuroimaging. We have tried to introduce some of the novel methods in the introduction in a language that is more suitable to gain the attention of a larger public. We think that in a methods paper it is appropriate to introduce the basic principles of new methods in the introduction rather than in the methods section itself. 

We discuss the imaging methods in use in higher detail in the supporting text S1 and S2. 

We have extended a bit on the Flat-top lag-windows method in the main paper that is presented in the “Bootstrapping on the time course level” section and added:

“The flat-top lag-windows method involves choosing window sizes that are large enough to capture the autocorrelation structure of the time series, but small enough to avoid “averaging” over too many observations.”

You rightfully pointed out that FDR corrections are somehow standard in imaging. As for the FDR correction we have added some sentences in the “NHST based image reproducibility” section of the methods and added: 

“All paths met the critical Bonferroni corrected p value even after conjunction analysis and stringent correction procedures, rendering more liberal methods like FDR corrections unnecessary. The Bonferroni thresholded data in our experiment automatically satisfies FDR correction criteria.”

Data Visualization: There is no mention of data visualization in the paper. Visualization tools can aid reader understanding of research findings. Therefore, incorporating charts or other visual elements to showcase key findings is recommended. Specifically, presenting statistical differences in brain network maps for a certain disease, such as ADHD (DOI: https://doi.org/10.1016/j.knosys.2022.109082) and PD (DOI: 10.1109/TCBB.2023.3252577), compared to a normal control group would be beneficial.

As for visualization: We have now added a quotation in the paper of a software package that was used to visualize the connectomes that was developed by the Institute of Automation, Chinese Academy of Sciences. The package can be found at https://code.google.com/archive/p/visualconnectome/

As for visual elements: We have added an extra figure (now called figure 2) to the main paper that visualizes the four forms of connectivity including: connectivity, connectivity corrected for AR(1), Connectivity corrected for reliability, connectivity corrected for AR(1) and reliability. We hope that this figure summarizes all our findings comprehensively. 

We are not quite sure if the reviewers saw all our figures that were uploaded together with the paper. Maybe something went wrong when downloading the paper from the internet. In total five figures were generated that in all cases consisted of at least three panels. In total roughly 30 visual elements were created for the main paper while a larger number of figures consisting of multiple panels are present in the supporting materials. 

As for diseases: We have mentioned the papers in the limitation section as they may open a window to novel ways of analysis. In which we state:

The Monte Carlo simulation clearly showed that a connectome that was based on hypothetical brain activation maxima derived from meta-analysis yields far more reproducible results compared to connectomes that have been selected randomly. However, this does not imply that meta-analysis derived connectomes are the best possible solution. As an alternative one could derive connectomes from hypothesis free methods such as independent component analysis, clustering and deep factor learning [31–33]. It was beyond the reach of this study to prove that connectomes derived from resting state and structural data by means of hypothesis free methods outperform brain regions that have been based on the neural correlates of working memory.

We acknowledge that our analysis is restricted to healthy individuals. And we report in the limitation section of the paper the following:

It is obvious that our data are limited to healthy individuals so that we cannot make any conclusions about clinical populations that may exhibit different within subject time course reliability.

However, there are no data from diseased populations that have been assessed using the same scanner parameters and working memory experiment as in our study. 

Consistent Terminology Usage: Throughout the paper, consistency in the usage of concepts and terminology should be maintained to avoid reader confusion. For instance, if "within-subject time course reliability" is used in the text, it should be consistently utilized throughout the paper without substitution.

The reviewer correctly pointed out that we are somewhat inconsistent with the phrase "within-subject time course reliability". We have now corrected this issue. However, in a few cases we use the word time course reliability when the phrase within-subject time course reliability was used in the immediate context of the sentence to enhance readability. 

Critical Reflection on Methods: When describing research methods, incorporating discussion on method limitations, such as potential biases in data collection or issues encountered in statistical analysis, can enhance the paper's comprehensiveness and reader understanding.

As for data collection: We have tried to raise a sample that is representative for the Austrian general population demographics. Moreover, we have applied rather strict selection criteria with regard to task performance and head motion so that the sample was substantially reduced. 

As for limitations: we would like to point out that in the original paper several limitations were mentioned. We have now added some extra limitations and given the limitations section its own sub heading. 

As for problems that were encountered in the data analysis: We in fact discussed in the supporting text S1 and S2 problems that were encountered. In these sections we reflect on the nature of negative reliability and its relation to auto correlation structure. We also reflect on the problems that we encountered when we had to deal with negative reliability in the context of averaging procedures. We have now added an additional analysis that reflects on the accuracy of AR(1) corrections. 

Reviewer #2: This study investigates the influence of within-subject time course reliability on the detection of connectivity and group effects using NHST, while accounting for variables such as sample size and spatial distribution of connectome nodes. The study performs a number of detailed analyses to tease apart the impact of these factors on the reliability and interpretability of functional connectivity measures in neuroimaging research. The paper is well written and makes an important point. To enhance clarity and the study's impact, I propose significant revisions to elaborate on these aspects. I have a few suggestions that could further enhance the clarity and impact of the study.

Major:

1. The introduction section would benefit from a more explicit statement of the paper's objectives. While the aim to investigate the impact of within-subject time course reliability on connectivity detection and group effects is implied, a clearer articulation of the specific research questions or hypotheses would enhance reader understanding.

We have substantially extended the introduction in the hope that the research hypotheses are clearly formulated. As reported above we have now defined the four-research hypothesis in greater detail at the end of the introduction that reads as:

Here we test four hypotheses regarding the detectability of connectivity. We conducted a working memory experiment on two separate occasions and extracted two working memory-related functional brain networks. These networks were based on brain activity maxima identified in a previous large-scale meta-analysis that examined the neural correlates of working memory [20]. First, we investigated the effect of within-subject time course reliability on detectable connectivity. We hypothesized that connectivity among regions will be higher than the within-subject time course reliability. When observed connectivity exceeds the precision of time course measurement, we replace the connectivity correlation with the within-subject time course reliability correlation and refer to this as detectable connectivity. Second, we investigated the effect of within-subject time course reliability on null hypothesis significance testing (NHST) based group reproducibility. We expected that the mean detectable connectivity may be substantially lower than mainstream connectivity. Therefore, we hypothesized that mainstream group connectivity statistics obtained by testing connectivity correlations against zero are likely inflated. Third, we investigated the effects of sample size on group reproducibility. It is well established that power to detect group effects increases with larger sample sizes. We hypothesized that the differences between group statistics based on observed versus detectable connectivity will decrease with larger samples. Fourth, we investigated the effect of connectome selection on group reproducibility. It is reasonable to assume that a hypothetical connectome based on working memory-related brain coordinates may contain more robust brain activity data and therefore yield more reliable outcomes compared to randomly generated connectomes.

2. Additionally, the introduction occasionally suffers from a dense presentation of technical terms and concepts, which could potentially alienate readers unfamiliar with the intricacies of fMRI methodology.

We have tried to rewrite the introduction such that is more readable for a larger audience. We have tried to introduce the statistical concepts more carefully in the hope that it satisfies the reviewer and the audience. Key concepts that are needed to guide the reader trough the manuscript are printed in a cursive font after which they are defined. 

3. Some of the methods suffer from slightly extreme language which is easily falsifiable, one can never ‘rule out the possibility” that “any lack” was due to poor performance or motion. Pease rework the text to be more balanced throughout.

We have reworked the text, as suggested. 

4. The abstract mentioned 0.81 percent (is this correct, less than 1%?)…also where are the results that show this? It was not obvious from the paper.

In fact, the rationale was given in the beginning of the discussion where we stated that: “When corrected for residual auto correlations, the detectable amount of connectivity decreased even further to 0.09. Taking the square root of 0.09, we can infer that only 0.081 percent of the observed connectivity might be attributed to cognitive-related signal fluctuations”. We concur with the reviewer's assessment that the rationale was not adequately presented in the key section of the paper, specifically the abstract. Consequently, we have revised the abstract to enhance the clarity of the rationale behind this “slightly” shocking news. The key sentence in the abstract reads as follows: “The grand mean connectivity of the connectome equaled r =0.41 (95% CI 0.31-0.50) for the test run and r =0.40 (95% CI 0.29-0.49) for the retest run. The mean connectivity decreased to r = 0.09 (95% CI 0.03-0.16) when test-retest reliability and auto-correlations of single time courses were considered, indicating that less than a quarter of connectivity is detectable. The square root of the detectable connectivity r = 0.09 suggests that only 0.81% of the connectivity is explained by working memory-related time course fluctuations.”

5. Figures could be improved (e.g. figure on, needs legends for blue/reg plots for all panels)

We have added the legends.

6. Could the authors also plot the time course activity task effects? So not just the task-modulated connectivity, as these are measuring different things.

We are unsure of the reviewer's intended meaning. Figure 1D displays the grand mean fMRI time course, clearly indicating 24 signal changes that are associated with the 24 cognitive events being studied.

7. While the Discussion section effectively addresses the impact of within-subject time course reliability on connectivity measures, could the authors provide additional insights into potential strategies for mitigating the effects of variability in fMRI studies? For example, are there specific methodological approaches or analytical techniques that could help account for individual differences in time course reliability and improve the robustness of connectivity analyses?

As for individual differences we have now added the following passage:

We observed that very large individual differences in within subject response time reliability exist (range =0.01-0.87). These individual differences in response time reliability were correlated (r~0.45) with individual differences in within subject time course reliability. This finding corroborates previous observations that cognitive aspects of tasks are related to within-subject time course reliability [1]. 

As for improving robustness, we have just finished a paper that shows that within subject time course reliability can be doubled if low and high pass filters are carefully designed for a specific experiment. We have added a passage in the current paper in which we discuss several approaches that might work including filter optimization. We have added following passage to the paper: 

What can be done to enhance within-subject time course reliability? A promising approach to improve within-subject time course reliability is to simply average task-related time courses obtained from multiple independent fMRI sessions. Alternatively, one might incorporate several time courses of the same individual that measure the same task into a multivariate statistical model such as structural equation modeling. Such multivariate techniques have been shown to improve reliability on the group level [5]. A cheaper alternative might be to refine high and low-pass filters used to separate noise from signal. Current fMRI software packages typically employ a one-size-fits-all filter approach, which may not be optimal. Theoretically, designing filters based on the expected shape of the task-related response could effectively separate noise from signal without excessively altering the autocorrelational structure of the time courses. Possibly ultra-high field scanners with more favorable signal to noise ratios could detect higher levels of within-subject time course reliability. Within this context, it would be beneficial to assess the contributions of signal-noise ratio under different measurement conditions and post-processing techniques through a sensitivity analysis. 

8. Additionally, considering the limitations discussed, do the authors have any recommendations for future research aimed at addressing these challenges and further validating these findings across diverse experiments. It may be beneficial to explore the potential

---

## [Decision Letter · Decision Letter 1]

28 Oct 2024

When most fMRI connectivity cannot be detected: insights from time course reliability

PONE-D-24-04077R1

Dear Dr. Koten,

We’re pleased to inform you that your manuscript has been judged scientifically suitable for publication and will be formally accepted for publication once it meets all outstanding technical requirements.

Kind regards,

Federico Giove, PhD

Academic Editor

PLOS ONE

Additional Editor Comments (optional):

Reviewers' comments:

Reviewer's Responses to Questions

**Comments to the Author**

1. If the authors have adequately addressed your comments raised in a previous round of review and you feel that this manuscript is now acceptable for publication, you may indicate that here to bypass the “Comments to the Author” section, enter your conflict of interest statement in the “Confidential to Editor” section, and submit your "Accept" recommendation.

Reviewer #1: All comments have been addressed

2. Is the manuscript technically sound, and do the data support the conclusions?

Reviewer #1: Yes

3. Has the statistical analysis been performed appropriately and rigorously? 

Reviewer #1: No

4. Have the authors made all data underlying the findings in their manuscript fully available?

Reviewer #1: Yes

5. Is the manuscript presented in an intelligible fashion and written in standard English?

Reviewer #1: Yes

6. Review Comments to the Author

Reviewer #1: Although the author addressed most of the issues, the explanation regarding FDR correction did not convince me.

7. PLOS authors have the option to publish the peer review history of their article (what does this mean?). If published, this will include your full peer review and any attached files.

Reviewer #1: No

---

## [Editor Report · Acceptance letter]

4 Dec 2024

PONE-D-24-04077R1 

PLOS ONE

Dear Dr. Koten, 

I'm pleased to inform you that your manuscript has been deemed suitable for publication in PLOS ONE. Congratulations! Your manuscript is now being handed over to our production team.

Kind regards, 

on behalf of

Dr. Federico Giove 

Academic Editor

PLOS ONE